# ALL IN ONE: UNIFIED PRETRAINING OF GUI AGENTS VIA MASKED TRAJECTORY PREDICTION

## ABSTRACT

Graphical User Interface (GUI) agents are intelligent systems that interact with software applications by perceiving visual elements and taking appropriate actions. Existing studies typically explore a wide range of pretraining strategies with heterogeneous corpora and directly unify these tasks through mixture training to enhance the generalization of GUI agents. However, the direct unification of existing pretraining strategies leads to inconsistent training objectives and data heterogeneity, preventing the full potential of each pretraining task from being realized. In this paper, we present a unified framework, **M**asked **T**rajectory **P**rediction (MTP), which consolidates diverse pretraining strategies into a consistent training objective via a masking-based manner. Specifically, we collect open-source GUI corpora that encompass a broad range of logical and semantic coherence, including randomly generated action–screenshot pairs, GUI tutorial data, and human-annotated datasets. Then, MTP models each GUI multi-interaction as a trajectory and defines pretraining objectives through component masking and prediction. Furthermore, to handle the heterogeneity across open-source corpora, we design a role-aware adapter learning module that dynamically routes each token to an appropriate optimization path. Extensive experiments on four representative GUI navigation benchmarks (AndroidControl, GUI-Odyssey, AITZ, and Mind2Web) demonstrate the effectiveness and generalization ability of our framework. By unifying existing pretraining objectives, MTP significantly outperforms prior methods and achieves SOTA results. The code and dataset will be publicly released.

## 1 INTRODUCTION

Intelligent agents are autonomous systems that perceive environments and imitate humans to take actions toward specific goals (Wang et al., 2024a). As a vital subset, graphical user interface (GUI) agents interpret visual elements, reason over user instructions, and engage in multi-turn interactions within GUI environments (Wang et al., 2024c). Enabled by the recent progress of large vision-language models (LVLMs), GUI agents have begun to adopt them as foundation models (Wang et al., 2024b; Yue et al, 2025; Chen et al., 2024c), followed by pretraining on GUI-specific corpora. With these advancements, GUI agents are becoming versatile personal assistants that support tasks ranging from online shopping to productivity tool operation (Rawles et al., 2023).

Building upon prior GUI studies, we broadly categorize existing pretraining efforts into three paradigms: 1). State transition prediction (Shen et al., 2024; Sun et al., 2024; Qin et al., 2025) tasks GUI agents to predict the intermediate user action and explain its underlying rationale when given two consecutive interface states. 2). Action-centric chain-of-thought (CoT) prediction (Xu et al., 2024; Huang et al., 2025; Xie et al., 2025; Lin et al., 2025) provides the current screenshot and user instruction, asking GUI agents to produce a structured reasoning trace. 3). Multi-turn prediction (Zhang et al., 2025b; Qin et al., 2025; Zhang et al., 2025a) formulates GUI tasks as dialogue-like interactions between users and GUI devices, requiring agents to process sequential visual inputs and action histories.

While these paradigms have advanced GUI agents in different directions, each remains focused on isolated abilities, such as local state transitions, reasoning traces, or multi-turn interactions, which ultimately hinders the development of generalizable agents. Recent state-of-the-art (SOTA) stud-

Figure 1: **Left:** Direct unified mixture training vs. MTP on the AITZ benchmark (Zhang et al., 2024). **Right:** An overview of MTP. By masking arbitrary trajectory contents and predicting the counterparts, MTP unifies heterogeneous GUI corpora under a consistent pretraining objective.

ies (Qin et al., 2025; Zhang et al., 2025b; Wang et al., 2025) are therefore to consolidate existing tasks and corpora into a unified mixture to leverage complementary strengths and enable broader applicability in real-world scenarios. Nevertheless, the direct unified mixture training fails to yield performance improvements, and as shown in Figure 1 Left, the direct unified pretraining of action-centric CoT prediction and state transition prediction even underperforms relative to their single-task counterparts. This phenomenon reflects two major challenges faced by the unified process. 1). **Inconsistent training objectives**. The three existing paradigms each capture distinct local marginal distributions of GUI interactions, which correspond to different partial dependencies among states, reasoning traces, and partial interaction histories. Specifically, state transition prediction focuses on the local association between two consecutive states and the corresponding action. Action-centric chain-of-thought prediction captures only the local alignment between the instruction and a reasoning trace. Multi-turn prediction is restricted to partial historical states and lacks the constraining feedback necessary from future outcomes. 2). **Data heterogeneity.** The three existing paradigms collect substantial data through both automated pipelines and manual annotation, yet the resulting corpora are divided into task-specific and heterogeneous formats that hinder integration. For instance, such heterogeneity appears in reasoning steps that differ in length or lack explicit processes, such as screen observations, posing additional challenges for unified modeling.

To break the boundary, we propose a unified yet flexible framework that enforces a consistent training objective across diverse GUI pretraining tasks, as shown in Figure 1 Right. Specifically, we formalize the pretraining task as a mask-based trajectory prediction problem, where each GUI corpus is treated as a trajectory. Each trajectory is a multi-modal sequence interleaved by user interfaces, diverse reasoning paths, and actions. During training, one or more components in the trajectory are substituted by the [Mask] token, and the GUI agent is required to predict these components in an auto-regressive manner. By casting different paradigms into the same trajectory masking formulation, MTP transforms the optimization from local marginal dependencies to a unified objective over complete trajectories, thus resolving the inconsistency among previous training objectives. As illustrated in Figure 1 Left, MTP unifies all three existing pretraining paradigms and outperforms their pairwise combinations, validating that MTP scales with unified training.

Beyond the task formulation, we also introduce a role-aware adapter learning module to tackle data heterogeneity, which dynamically routes each token from different corpora to an appropriate optimization path based on its semantic role. Benefiting from this unified framework, we systematically organize open-source GUI corpora and conduct large-scale GUI pretraining. The corpora span a range of logical and semantic coherence, covering randomly generated action–screenshot pairs (Wu et al., 2024a), GUI tutorial data (Zhang et al., 2025a; Jang et al., 2025; Sun et al., 2025; Chen et al., 2024b), and human-annotated datasets (Xu et al., 2024; Chai et al., 2024). Extensive experiments across multiple public benchmarks demonstrate the effectiveness of the framework, with MTP achieving significant improvements over existing methods and marking a considerable advancement in GUI agent research.

The main contributions of our work are threefold:

- Through a systematic analysis of existing GUI pretraining paradigms, we reveal an overlooked bottleneck where inconsistent objectives and heterogeneous corpora hinder the unified scaling of GUI pretraining.

- To break this boundary, we propose MTP, a unified yet flexible pretraining framework that establishes a consistent training objective for GUI agents through masked trajectory prediction and a role-aware adapter learning module.

- We conduct comprehensive experiments on four representative GUI navigation datasets, AndroidControl, GUI-Odyssey, AITZ, Mind2Web. The results verify the effectiveness and generalization of MTP, revealing a potential direction for future GUI agent research.

## 2 RELATED WORK

### 2.1 GUI AGENT

The advancements in Large Language Models (LLMs) and LVLMs have significantly accelerated the development of GUI agents (Wang et al., 2024b; Bai et al., 2025; OpenAI, 2024). Early attempts at developing GUI agents involve parsing GUIs into source code and feeding them into LLMs for action inference (Shi et al., 2017; Kim et al, 2023; Chen et al., 2024a). However, these approaches rely on internal APIs or system backend code, which are typically inaccessible in commercial software. This limitation has prompted a shift in research toward purely vision-based agents.

Recently, GUI agents, such as UI-TARS series (Qin et al., 2025; Wang et al., 2025), Show-UI (Qin et al., 2025), and CogAgent (Hong et al., 2024), have employed LVLMs as the foundational model to predict actions conditioned on GUI screenshots. These works focus on empowering two core abilities in GUI agents: interpreting GUI contexts and imitating human actions by following user instructions. SeeClick (Cheng et al., 2024) is one of the pioneering approaches to tackle GUI grounding from a pure visual perspective, and it further contributes an automated pipeline for constructing grounding data. Additionally, GUI-R1 (Luo et al., 2025) designs various reward functions, such as those for clicking icons, to further enhance the model's ability to interpret visual elements in GUI environments. Regarding the imitation of human actions by following user instructions, subsequent work (Zhang et al., 2025a; Xu et al., 2024) has utilized human-annotated GUI agent data and various instructional tutorials to enhance agents' planning capabilities.

### 2.2 GUI CORPUS

GUI corpora commonly encompass diverse user interactions ranging from social media, map navigation, to online shopping, and span various platforms such as mobile, web, and desktop. Android in The Wild (Rawles et al., 2023) is the first to introduce a 715k-episode GUI sequence from Android devices, featuring 30k unique instructions and screenshots with varying resolutions. Following this, Android in the Zoo (Zhang et al., 2024) extends the dataset with CoT annotations to stimulate the reasoning process prior to taking actions. To broaden the application scope, GUI-Odyssey (Lu et al., 2024) focuses on cross-app navigation to emulate real user experiences in communication, entertainment, and productivity scenarios.

Although multiple works have introduced high-quality GUI corpora, the limited scale of manually annotated data remains insufficient for training LVLMs (Wang et al., 2024b; Bai et al., 2025). To address this, existing pretraining efforts (Zhang et al., 2025a; Jang et al., 2025; Shen et al., 2024; Wu et al., 2024a) commonly collect GUI data from web-based and instructional videos, and generate random action–screenshot trajectories to support large-scale training. Specifically, Mobile3M (Wu et al., 2024a) initiates this direction by pretraining on 20 million synthetic interactions randomly sampled from 49 popular Chinese apps. Building on this, MONDAY (Jang et al., 2025) introduces a more realistic corpus with 313k annotated frames from 20k instructional videos, capturing diverse user behaviors in real-world mobile navigation.

## 3 METHOD

### 3.1 OVERVIEW

In this section, we describe our proposed framework MTP, which consists of three core components: 1). The UniTraj dataset, a large-scale collection of GUI interaction sequences collected from multiple open-source corpora. 2). Mask trajectory prediction task, which treats each GUI sequence

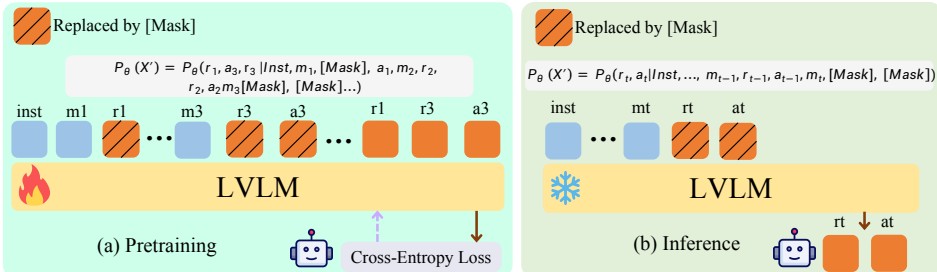

Figure 2: Overview of the MTP paradigm during pretraining and inference.

as a trajectory and requires models to predict its arbitrary masked component. 3). The role-aware adapter learning module, which incorporates token-wise adapters to effectively handle divergent optimization directions introduced from heterogeneous GUI corpora.

## 3.2 UNITRAJ DATASET

Based on a comprehensive analysis of existing pretraining corpora, we introduce UniTraj, a large-scale unified trajectory dataset that standardizes cross-corpora user interactions. UniTraj integrates major public datasets from diverse pretraining paradigms (Zhang et al., 2025a; Xu et al., 2024; Wu et al., 2024a; Jang et al., 2025; Sun et al., 2024; 2025), establishing a unified action space encompassing eleven action types applicable across five major operating systems, including Windows, macOS, Android, iOS, and Linux. Details of the action types can be found in the Appendix A. The average length of obtained trajectories is 5.6 steps, capturing complex, multi-step user interactions and the corresponding reasoning processes.

Specifically, we characterize UniTraj's samples through two coherence levels: logical coherent trajectories and semantically coherent trajectories. Logical coherent trajectories exhibit a consistent flow of action–screenshot pairs but lack corresponding instruction annotations, and thus do not reflect goal-directed action. They constitute 85.4% (416K) of the dataset, sourced primarily from Falcon-UI and OS-Genesis (Shen et al., 2024; Sun et al., 2024), which consists of randomly generated action sequences paired with corresponding GUI screenshots. In contrast, semantically coherent trajectories are aligned with a clearly defined high-level task, with each action contributing to task completion while also reflecting a CoT rationale behind its selection. They constitute 14.6% (71K) of the dataset, sourced from web tutorials (Zhang et al., 2025a), instructional videos (Jang et al., 2025), and human-annotated GUI datasets (Xu et al., 2024).

Since these trajectories originate from different pretraining schemes (Xu et al., 2024; Zhang et al., 2025a; Jang et al., 2025; Wu et al., 2024a), they exhibit inherent heterogeneity. For instance, CoT annotations within these trajectories vary in structure, with some containing observations, intermediate reasoning steps, and low-level instructions, while others omit certain components. Further illustrative samples are presented in the Appendix B.

## 3.3 MASK TRAJECTORY PREDICTION

In this section, we present the core insight of our pretraining framework, which establishes a consistent training objective that can accommodate heterogeneous GUI corpora. To achieve this, we regard the multi-turn interaction between users and GUI interfaces as a multi-modal trajectory, which may contain a user instruction, a sequence of screenshots, actions, and the associated reasoning processes. Notably, under this structure, applying a mask at any position within the trajectory, such as on actions or components of the reasoning steps, can effectively align diverse pretraining tasks under the same objective.

The core idea of masked trajectory prediction is to mask a fixed proportion of each component in the trajectory and guide the GUI agent to predict the missing parts. Specifically, this strategy enables the GUI agent to learn effectively even in the absence of partial contextual information, thereby enhancing its robustness and ability to generalize across diverse scenarios.

Formally, given a GUI agent $\theta$ and a semantically coherent GUI trajectory $T = (\text{Inst}, \{(m_t, r_t, a_t)\}_{t=1}^T)$, we randomly mask a fixed proportion of components within the trajec-

Figure 3: Illustration of the role-aware adapter learning module. Given a masked trajectory with heterogeneous components, the module dynamically assigns each token to a specific adapter for targeted optimization. Abbreviation: H Inst=High level instruction; L Inst=Low level instruction; Scne=Screenshot; Tht=Thought; Obv=Observation; Act=Action.

tory. As exemplified in Figure 1 Right, MTP replaces $r_1$ (CoT reasoning of the $1^{st}$ step), $r_3$ (CoT reasoning of the $3^{rd}$ step), $a_3$ (action of the $3^{rd}$ step) in the trajectory $T$ with the special $[\texttt{Mask}]$ token, resulting in a masked version denoted as $X'$. As illustrated in Figure 2(a), the pretraining objective is to predict the masked components based on the masked trajectory $X'$, which can be formulated as the following conditional probability:

$$P_\theta(X') = P_\theta\big(r_1, a_3, r_3 \mid Inst, m_1, [\texttt{Mask}], a_1, m_2, r_2, a_2, m_3, [\texttt{Mask}], [\texttt{Mask}], \dots\big). \quad (1)$$

Note that our mask trajectory prediction strategy can be applied to any GUI trajectories, regardless of whether they exhibit semantic coherence or logical coherence. In the case of logically coherent trajectories, the objective of MTP can be formulated as the following conditional probability:

$$P_\theta(X') = P_\theta\big(r_1, a_3, r_3 \mid m_1, [\texttt{Mask}], a_1, m_2, r_2, a_2, m_3, [\texttt{Mask}], [\texttt{Mask}], \dots\big). \quad (2)$$

Inspired by the success of masked autoencoders in the vision domain (He et al., 2022), we adopt a relatively high masked ratio to increase task difficulty and encourage the GUI agent to reason over long-horizon dependencies within GUI trajectories. Empirically, we find that masking 80% of the components within a trajectory achieves the best trade-off between task difficulty and model performance. Considering the noisy action-centric CoT annotations and low-quality screenshots in existing open-source trajectory datasets, we avoid extreme masking ratios, such as 100%, which could adversely impact learning stability. An 80% masking ratio compels the GUI agent to predict the masked components from the masked trajectory, while still ensuring sufficient data utilization and effective supervision.

In the downstream inference stage, MTP simulates the same masking configuration as used during pretraining by replacing the current-step CoT and action with the special $[\texttt{Mask}]$ token. As illustrated in Figure 2(b), the GUI agent predicts the corresponding reasoning and action based on the contextual trajectory, and the inference objective can be expressed as follows:

$$P_\theta(X') = P_\theta\big(a_t, r_t \mid inst, \dots, m_{t-1}, r_{t-1}, a_{t-1}, m_t, [\texttt{Mask}], [\texttt{Mask}]\big). \quad (3)$$

where $a_t$ and $r_t$ denote the predicted action and the corresponding reasoning process at step $t$, respectively.

## 3.4 ROLE-AWARE ADAPTER LEARNING

In the previous section, we have discussed how MTP enables a consistent training objective across diverse pretraining tasks, but the substantial data heterogeneity across existing GUI corpora presents a significant challenge for unified modeling. In particular, our analysis of the UniTraj dataset, as illustrated in Figure 3, shows that trajectories can contain up to six components but often include only a subset. Furthermore, the quality of images varies significantly, with human-collected screenshots being generally high-quality, while those extracted from web tutorials often contain instructional visual elements such as red circles, arrows, or overlays (Zhang et al., 2025a).

Accordingly, we introduce a role-aware adapter learning module to address the challenge of data heterogeneity. Since prior pretraining methods predominantly adopt Low-Rank Adaptation (LoRA)

training (Hu et al., 2022), we extend LoRA by introducing multiple specialized adapters and a token-wise router that dynamically selects one for each token based on its role in the GUI trajectory.

To begin with, we briefly review the core concepts of LoRA. It assumes that parameter updates lie in a low-dimensional subspace, allowing training to be performed through a low-rank decomposition while keeping the pretrained weights frozen. Based on this formulation, the forward pass of a LoRA layer can be expressed as follows:

$$\Delta W_0 = BA, \quad h = W_0 x + \alpha \cdot \Delta W_0 x \tag{4}$$

where $x \in R^k$ is the input feature, $W_0 \in R^{d \times k}$ denotes the frozen pretrained weight, and $\Delta W_0$ is the trainable update parameterized by a low-rank decomposition, with $B \in R^{d \times r}$ and $A \in R^{r \times k}$, such that $r \ll \min(d, k)$. The scalar $\alpha$ controls the contribution of the update during training.

As illustrated in Figure 3, this module extends standard LoRA by introducing multiple adapters for each component in the GUI trajectory, aiming to address the data heterogeneity inherent in existing pretraining corpora. Specifically, to dynamically assign different tokens to appropriate adapters, a token-wise routing mechanism is employed. It selects the most suitable adapter for each token based on a linear scoring function:

$$\hat{i} = \arg\max_i \left( w_i^\top x \right), \quad \Delta W_0 = B_{\hat{i}} A_{\hat{i}}, \quad h = W_0 x + \alpha \cdot \Delta W_0 x \tag{5}$$

where $x$ is the input feature, $w_i$ is the learnable routing weight for the $i$-th adapter, and $B_i, A_i$ are the low-rank matrices associated with adapter $i$. The selected adapter $\hat{i}$ is used to generate the low-rank update $\Delta W_0$, which is then applied during forward propagation.

## 4 EXPERIMENTS

In this section, we validate the effectiveness of MTP on comprehensive GUI agent benchmarks. Specifically, we adopt Qwen2-VL-2B (Wang et al., 2024b), Qwen2.5-VL-3B (Bai et al., 2025), Qwen2.5-VL-7B (Wang et al., 2024b), and MiMo-VL (Yue et al, 2025) as base models, and leverage the open-source UniTraj dataset for training. Implementation details are given in the Appendix C.

### 4.1 DATASETS AND EVALUATION METRICS

We evaluate the multi-step execution capabilities of MTP using four navigation datasets, including AndroidControl (Li et al., 2024), GUI-Odyssey (Lu et al., 2024), AITZ (Zhang et al., 2024), and Mind2Web (Deng et al., 2023). The detailed statistics of these datasets can be found in the Appendix E. Based on the presence of high-level versus low-level instructions, we categorize the datasets into two groups. The first group, comprising datasets such as AndroidControl-High, GUI-Odyssey, and Mind2Web, evaluates the agent's capability for global or long-horizon planning. The second group includes datasets like AndroidControl-Low, which focus on fine-grained action execution at the step level.

For evaluation metrics, we adopt three standard metrics commonly used in GUI agent bench-marks, denoted as Type, Grounding, and SR, respectively (Wu et al., 2024b). Specifically, Type denotes whether the predicted action type exactly matches the ground truth. Grounding evaluates the accuracy of coordinate prediction for click actions, reflecting the GUI agent's grounding capability. Step-wise Success Rate (SR) quantifies whether both the action type and its associated arguments are completely correct at each step.

Table 1: Analysis of unified pretraining: MTP vs. direct unified mixture training. Abbreviation: STP=state transition prediction; ACP=action-centric CoT prediction; MP=multi-turn prediction; Direct=direct unified mixture training.

| Paradigm | Scale | AndroidControl-High | | |
|---|---|---|---|---|
| | | Step Acc | Ground Acc | Step Acc |
| – | – | 84.00 | 66.00 | 62.50 |
| Individual | | | | |
| STP | 416K | 84.86 | 69.48 | 65.77 |
| ACP | 34K | 85.39 | 68.60 | 65.57 |
| MP | 37K | 84.97 | 68.38 | 64.95 |
| ACP + MP Unification | | | | |
| Direct | 71K | 85.04 | 68.45 | 65.22 |
| MTP | 71K | 85.32 | 69.23 | 65.99 |
| STP + ACP + MP Unification | | | | |
| Direct | 487K | 85.28 | 69.60 | 66.12 |
| MTP | 487K | 85.35 | 70.48 | 66.55 |

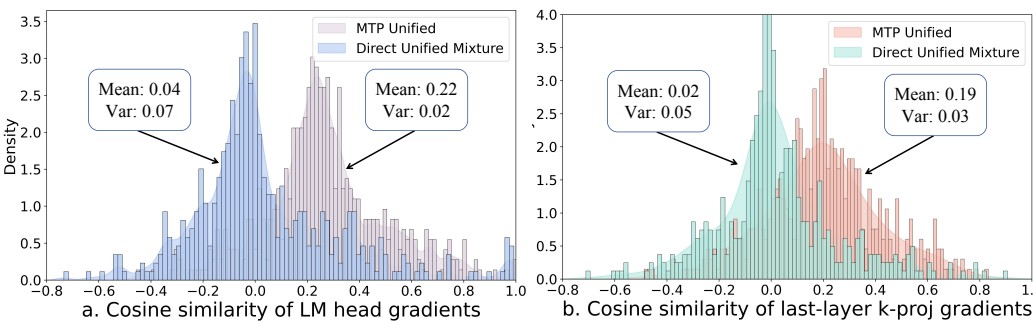

Figure 4: Kernel density estimation plots of gradient cosine similarity

Table 2: Results on AndroidControl (Li et al., 2024). AndroidControl-Low refers to the scenario where both low-level and high-level instructions are provided as inputs, while AndroidControl-High indicates that only high-level instructions are given. The best GUI agent is in **bold**, and the second rank is in underlined.

| Methods | Param. | AndroidControl-High | | | AndroidControl-Low | | |
|---|---|---|---|---|---|---|---|
| | | Type | Grounding | SR | Type | Grounding | SR |
| Claude (Anthropic, 2024) | – | 63.7 | 0.0 | 12.5 | 74.3 | 0.0 | 19.4 |
| GPT-4o (OpenAI, 2024) | – | 66.3 | 0.0 | 20.8 | 74.3 | 0.0 | 19.4 |
| Falcon-UI (Shen et al., 2024) | 7B | – | – | 72.7 | – | – | 86.6 |
| SeeClick (Cheng et al., 2024) | 9.6B | 82.9 | 62.9 | 59.1 | 93.0 | 73.4 | 75.0 |
| OS-Atlas (Wu et al., 2024b) | 7B | 85.2 | 78.5 | 71.2 | 93.6 | 88.0 | 85.2 |
| Aguvis (Xu et al., 2024) | 7B | – | – | 61.5 | – | – | 80.5 |
| UI-TARS (Qin et al., 2025) | 7B | 83.7 | **80.5** | 72.5 | 98.0 | **89.3** | **90.8** |
| MiMo-VL (Yue et al, 2025) | 7B | 87.1 | 74.0 | 70.1 | 98.0 | 85.5 | 87.9 |
| MiMo-VL + Ours | 7B | $87.8_{+0.7}$ | $75.6_{+1.6}$ | $72.0_{+1.9}$ | $98.1_{+0.1}$ | $86.9_{+1.4}$ | $88.6_{+0.7}$ |
| Qwen2-VL (Wang et al., 2024b) | 2B | 84.0 | 66.0 | 62.5 | 97.8 | 84.0 | 86.3 |
| Qwen2-VL + Ours | 2B | $85.3_{+1.3}$ | $70.5_{+4.5}$ | $66.5_{+4.0}$ | $98.0_{+0.2}$ | $86.3_{+2.3}$ | $88.0_{+1.7}$ |
| Qwen2.5VL (Bai et al., 2025) | 3B | 85.9 | 73.6 | 68.2 | 97.9 | 87.3 | 88.5 |
| Qwen2.5VL + Ours | 3B | $86.4_{+0.5}$ | $74.4_{+0.8}$ | $69.7_{+1.5}$ | $98.1_{+0.2}$ | $88.0_{+0.7}$ | $89.4_{+0.9}$ |
| Qwen2.5VL (Bai et al., 2025) | 7B | 87.9 | 76.4 | 72.9 | 98.1 | 87.3 | 88.2 |
| Qwen2.5VL + Ours | 7B | $\mathbf{88.8}_{+0.9}$ | $78.0_{+1.6}$ | $\mathbf{74.4}_{+1.5}$ | $\mathbf{98.3}_{+0.2}$ | $88.0_{+0.7}$ | $89.5_{+1.3}$ |

## 4.2 SUPERIORITY OF MTP IN UNIFIED PRETRAINING

To investigate effective unification strategies, we compare direct mixture training with MTP, drawing on both experimental evidence and theoretical insights. As illustrated in Table 1, we adopt Qwen2-VL-2B (Wang et al., 2024b) as the baseline and evaluate its performance under different pretraining strategies on the AndroidControl-High dataset. While individual pretraining on state transition, action-centric, and multi-turn prediction improves performance, directly mixing action-centric and multi-turn prediction performs worse than action-centric prediction alone. In contrast, MTP yields clear gains when combining action-centric prediction data with multi-turn prediction data, and further incorporating state transition data demonstrates its scalability with continued performance improvements.

To further investigate why MTP shows advantages in unification, we compare the cosine similarity of gradients between MTP and direct mixture training. As shown in Figure 4, kernel density estimation plots at the LM head and $k$-projection layer reveal that MTP yields a mean cosine similarity approximately 0.17 higher and a variance about 0.03 lower than direct mixture training. This higher cosine similarity reflects more consistent training objectives, while the lower variance suggests more stable training (Ciernik et al., 2024), both of which highlight MTP's superiority in unified optimizing training. Further theoretical analysis is provided in the Appendix F.

## 4.3 MAIN RESULT

As illustrated in the Table 2 and Table 3, we present a comprehensive comparison with existing GUI agent pretraining methods, and MTP consistently achieves superior performance across all evaluated

Table 3: Results on GUI-Odyssey (Lu et al., 2024), AITZ (Zhang et al., 2024), and Mind2Web (Deng et al., 2023) are reported, where AITZ uses SR as the evaluation metric, and Mind2Web reports overall performance across Cross-Web, Cross-Task, and Cross-Domain scenarios. The best GUI agent is in **bold**, and the second-best is underlined.

| Methods | Param. | GUI-Odyssey | | | AITZ | Mind2Web |
| --- | --- | --- | --- | --- | --- | --- |
| | | Type | Grounding | SR | | |
| Claude (Anthropic, 2024) | – | 60.9 | 0.0 | 3.1 | – | – |
| GPT-4o (OpenAI, 2024) | – | 34.3 | 0.0 | 3.3 | – | 56.6 |
| Falcon-UI (Shen et al., 2024) | 7B | – | – | – | 69.1 | 27.6 |
| SeeClick (Cheng et al., 2024) | 9.6B | 71.0 | 52.4 | 53.9 | – | 20.9 |
| OS-Atlas (Wu et al., 2024b) | 7B | 84.5 | 67.8 | 62.0 | – | – |
| Aguvis (Xu et al., 2024) | 7B | – | – | – | – | 57.2 |
| UI-TARS (Qin et al., 2025) | 7B | 94.6 | 90.1 | 87.0 | – | **63.1** |
| MiMo-VL (Yue et al, 2025) | 7B | 96.7 | 89.5 | 88.1 | 70.8 | 49.4 |
| MiMo-VL + Ours | 7B | $96.8_{+0.1}$ | $89.8_{+0.3}$ | $88.3_{+0.2}$ | $73.1_{+2.3}$ | $51.2_{+1.8}$ |
| Qwen2-VL (Wang et al., 2024b) | 2B | 96.1 | 86.1 | 84.8 | 66.6 | 46.7 |
| Qwen2-VL + Ours | 2B | $96.3_{+0.2}$ | $88.4_{+2.3}$ | $86.6_{+1.8}$ | $72.9_{+6.3}$ | $48.2_{+1.5}$ |
| Qwen2.5VL (Bai et al., 2025) | 3B | 96.1 | 88.1 | 87.2 | 71.3 | 54.5 |
| Qwen2.5VL + Ours | 3B | $\underline{97.5}_{+1.4}$ | $89.3_{+1.2}$ | $\underline{88.5}_{+1.3}$ | $\underline{73.2}_{+1.9}$ | $56.9_{+2.4}$ |
| Qwen2.5VL (Bai et al., 2025) | 7B | 96.2 | 88.2 | 86.3 | 72.1 | 57.1 |
| Qwen2.5VL + Ours | 7B | $\mathbf{97.9}_{+1.7}$ | $\mathbf{90.2}_{+2.0}$ | $\mathbf{88.7}_{+2.4}$ | $\mathbf{74.4}_{+2.3}$ | $\underline{59.3}_{+2.2}$ |

datasets. Multiple LVLMs (Wang et al., 2024b; Bai et al., 2025; Yue et al, 2025) are adopted as base models and trained on the open-source UniTraj dataset, where MTP consistently enhances their performance. We analyze the experimental results from global planning to local action execution.

### 4.3.1 GLOBAL PLANNING EVALUATION

For global planning evaluation, the GUI agent is required to predict the current action from a high-level instruction without access to low-level guidance. To illustrate the difference, consider a high-level instruction such as "Set an alarm for 7 AM," which requires multi-step navigation, versus a low-level instruction like "Tap on the '7' button," which provides single-step guidance.

Compared with existing GUI agent pretraining methods (Qin et al., 2025; Shen et al., 2024; Wu et al., 2024b; Cheng et al., 2024; Xu et al., 2024), MTP demonstrates superior performance on the AndroidControl-High, GUI-Odyssey, AITZ, and Mind2Web datasets. Among these, the AndroidControl-High dataset and the GUI-Odyssey dataset concentrate on the most prevalent mobile platforms for GUI agent evaluation. AndroidControl-High dataset (Li et al., 2024) covers various scenarios across 833 Android applications, while GUI-Odyssey (Lu et al., 2024) presents a long-horizon navigation challenge, with trajectories averaging 15.4 steps in length. As shown in Table 2 and Table 3, MTP on Qwen2.5-VL-7B achieves a higher SR than UI-TARS-7B (Qin et al., 2025), with improvements of 1.9% on AndroidControl-High and 1.7% on GUI-Odyssey, respectively. We attribute the performance gain to MTP's capacity to unify diverse pretraining tasks into consistent training objectives while effectively addressing the heterogeneity of GUI corpora.

Regarding evaluations beyond mobile platforms, MTP on Qwen2.5-VL-7B surpasses Falcon-UI (Shen et al., 2024) by 5.3% on the AITZ dataset (Zhang et al., 2024), covering diverse scenarios in Table 3. On the Mind2Web dataset (Deng et al., 2023), MTP on Qwen2.5-VL-7B achieves the second-best performance, narrowly trailing UI-TARS, which leverages extensive pretraining on 50B tokens of in-house data in Table 3. This superior performance stems from MTP's unified framework and role-aware adapter learning module, which jointly support consistent training across heterogeneous data from all prior pretraining strategies.

### 4.3.2 LOCAL PLANNING EVALUATION

For local planning evaluation, the GUI agent is provided with a current interface screenshot and a pair of high- and low-level instructions. Interestingly, although MTP is trained at the trajectory level, it still achieves competitive performance on the AndroidControl-Low dataset (Li et al., 2024).

Table 4: Ablation experiments about mask ratio on the AndroidControl-High dataset.

| Mask Ratio | Type | Grounding | SR |
|---|---|---|---|
| 0.2 | 85.87 | 73.16 | 68.62 |
| 0.5 | 86.12 | 73.95 | 69.30 |
| 0.8 | **86.42** | **74.36** | **69.71** |
| 1.0 | 86.10 | 73.84 | 69.12 |

Table 5: Ablation experiments about the number of adapters on the AndroidControl-High dataset.

| Num. Adapters | Type | Grounding | SR |
|---|---|---|---|
| 1 | 85.88 | 73.45 | 69.01 |
| 2 | 86.20 | 73.87 | 69.32 |
| 4 | **86.42** | 74.36 | **69.71** |
| 8 | 86.41 | **74.52** | 69.56 |

As shown in Table 2, MTP on Qwen2.5-VL-7B ranks second, slightly behind UI-TARS (Qin et al., 2025), which benefits from pretraining on 50B tokens, while outperforming Falcon-UI (Shen et al., 2024) by a margin of 2.9% on SR. We attribute this improvement to the inclusion of training data from various prior pretraining strategies, particularly those involving single-step instructions, which enhance the model's generalization to local planning tasks.

## 4.4 ABLATION STUDIES

### 4.4.1 ANALYSIS OF MTP GAINS ON MULTIPLE LVLMs

We adopt Qwen2-VL-2B (Wang et al., 2024b), Qwen2.5-VL-3B (Bai et al., 2025), Qwen2.5-VL-7B (Wang et al., 2024b), and MiMo-VL (Yue et al, 2025) as base models, and leverage the open-source UniTraj dataset for training. As shown in Table 2 and Table 3, MTP consistently enhances performance across a wide range of benchmarks, reflecting strong generalization across model scales and architectures. Moreover, it provides reliable improvements across different LVLMs, including SR gains of 3.1% on Qwen2-VL-2B for AndroidControl-High and 2.3% on MiMo-VL for AITZ. Notably, MTP is trained entirely on open-source data from the UniTraj dataset, which may have been partially observed during LVMLs' initial training.

### 4.4.2 ANALYSIS FOR MASK RATIO

Since the mask ratio is key to MTP, we investigate how different masking ratios (0.2, 0.5, 0.8, and 1.0) influence the model's effectiveness and conduct ablation studies on Qwen2.5-VL-3B (MTP). As shown in Table 4, performance consistently improves as the mask ratio increases from 0.2 to 0.8, while a significant drop is observed when the mask ratio reaches 1.0. Specifically, this upward trend can be attributed to more effective supervision, as higher mask ratios allow more data to contribute to gradients. While performance drops significantly when the mask ratio reaches 1.0, as the model is forced to rely solely on visual context from multiple screenshots to reason, thereby increasing training difficulty. Moreover, the presence of low-quality samples in the dataset, such as web-based GUI trajectory examples containing noisy visual cues like red circles (Zhang et al., 2025a), further increases the difficulty of learning and contributes to the observed performance degradation.

### 4.4.3 ANALYSIS FOR NUMBER OF ADAPTERS

To address the heterogeneity across diverse open-source pretraining GUI datasets, we propose a role-aware adapter learning module. This module dynamically assigns different components of a trajectory to specialized adapters, allowing targeted optimization with heterogeneous data. We investigate the impact of using 1, 2, 4, or 8 adapters on Qwen2.5-VL-3B (MTP) with AndroidControl-High dataset. As shown in Table 5, performance improves with more adapters, peaking at four, which outperforms the single-adapter LoRA baseline (Hu et al., 2022).

## 5 CONCLUSION

In this paper, we propose a unified framework, termed MTP, that consolidates diverse pretraining strategies into a single masked trajectory prediction formulation. By masking different components of GUI interaction trajectories and incorporating a role-aware adapter learning module, MTP unifies diverse pretraining objectives into a consistent objective and enables learning from heterogeneous GUI data. MTP has been validated on multiple LVLMs and outperforms existing GUI agents, such as UI-TARS, across multiple benchmarks.

## 6 ETHICS STATEMENT

All authors have read and commit to adhere to the ICLR Code of Ethics. We use publicly available or licensed datasets and apply anonymization or de-identification when needed. We evaluate model performance across demographic subgroups and discuss potential biases and limitations. We take responsibility for the work and will engage transparently should any ethical issues arise.

## 7 REPRODUCIBILITY STATEMENT

We have taken several measures to ensure the reproducibility of our work. The supplementary materials include detailed implementation code, covering model architectures and training procedures, that can be directly used to replicate our results. To substantiate our theoretical contributions, we provide full proofs and derivations of the proposed framework in Appendix F. Appendix A presents a comprehensive description of the pretraining datasets and their processing steps. Collectively, these materials enable other researchers to reproduce our findings and build on our work.

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

APPENDIX

The appendix includes the following aspects:

## A   UNITRAJ DATASET

Based on a comprehensive review of prior pretraining methods (Qin et al., 2025; Zhang et al., 2025a; Xu et al., 2024), we observe that the GUI corpora used in these studies can all be uniformly represented as trajectories. We have consolidated these datasets into a unified dataset, termed UniTraj. The UniTraj dataset encompasses five platforms and eleven action types, with an average trajectory length of 5.6 steps. The Table 6 presents a unified analysis of eleven action types.

The UniTraj dataset is categorized into two coherence levels: logical coherent trajectories and semantically coherent trajectories. Logical coherent trajectories exhibit a consistent flow of action–screenshot pairs but lack corresponding instruction annotations, and thus do not reflect goal-directed action. They constitute 87.7% (421K) of the dataset, sourced primarily from Falcon-UI (Shen et al., 2024) and OS-Genesis (Sun et al., 2024), which consists of randomly generated action sequences paired with corresponding GUI screenshots. In contrast, semantically coherent trajectories are aligned with a clearly defined high-level task, with each action contributing to task completion while also reflecting a CoT rationale behind its selection. They constitute 12.3% (59K) of the dataset, sourced from web tutorials (Zhang et al., 2025a), instructional videos (Jang et al., 2025), and human-annotated GUI datasets (Xu et al., 2024). The detailed open-source datasets are presented in Table 7.

## B   SUMMARIZATION OF EXISTING PARADIGMS

As illustrated in Figure 5, we summarize and categorize these strategies into three paradigms: 1). State transition prediction (Shen et al., 2024; Sun et al., 2024; Qin et al., 2025) tasks GUI agents to predict the intermediate user action and explain its underlying rationale when given two consecutive interface states. 2). Action-centric chain-of-thought (CoT) prediction (Xu et al., 2024; Huang et al.,

Table 6: Analysis of action types and corresponding descriptions in the UniTraj dataset.

| Action Space | Description |
|---|---|
| Tap | Tap on the specified position. |
| Type | Enter the specified text. |
| Swipe | Swipe on the screen. |
| OpenApp | Open the specified application. |
| LongPress | Long-press on the specified position. |
| Complete | No further actions required. |
| Incomplete | Requires additional steps. |
| Back | Return to the previous screen. |
| Home | Navigate to the home screen. |
| Wait | Temporarily pause the execution. |
| Enter | Confirm an input to the next step. |

Table 7: UniTraj Dataset Collection.

| UniTraj | Data Source | Trajs. |
|---|---|---|
| Logical Coherence | Mind2Web (Deng et al., 2023) | 1,009 |
| | GUIAct (Chen et al., 2024b) | 2,482 |
| | MiniWoB++ (min, 2023) | 2,762 |
| | AitZ (Zhang et al., 2024) | 1,987 |
| | AndroidControl (Li et al., 2024) | 13,594 |
| | GUI-Odyssey (Lu et al., 2024) | 7,735 |
| | AMEX (Chai et al., 2024) | 2,991 |
| | AitW (Rawles et al., 2023) | 2,346 |
| | TongUI (Zhang et al., 2025a) | 28,230 |
| | Monday (Jang et al., 2025) | 8,340 |
| | **Total** | **71,476** |
| Semantic Coherence | OS-Genesis (Sun et al., 2024) | 2,873 |
| | Falcon-UI (Shen et al., 2024) | 414,023 |
| | **Total** | **416,896** |

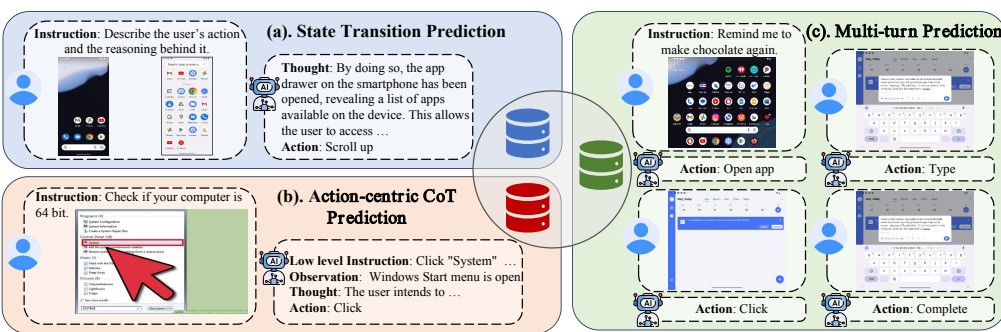

Figure 5: Comparison of pretraining paradigms for GUI agents. (a) State transition prediction focuses on describing state changes and the behind rationale. (b) Action-centric CoT prediction emphasizes step-by-step reasoning traces conditioned on the user instruction. (c) Multi-turn prediction involves sequential actions for complex tasks.

2025; Xie et al., 2025; Lin et al., 2025) provides the current screenshot and user instruction, asking GUI agents to produce a structured reasoning trace, comprising low-level instructions, corresponding observations, thoughts, and actions. 3). Multi-turn prediction (Zhang et al., 2025b; Qin et al., 2025; Zhang et al., 2025a) formulates GUI tasks as dialogue-like interactions between users and GUI devices, requiring agents to process sequential visual inputs and action histories.

## C  IMPLEMENTATION DETAILS

For MTP, which is built upon four LVLM backbones (Yue et al, 2025; Wang et al., 2024b; Bai et al., 2025), all input images are resized to 1280 × 720 to achieve a better trade-off between performance and efficiency. The maximum token sequence length is set to 8192 for each LVLM. During training, each LVLM in MTP is trained with a batch size of 32 and 8 gradient accumulation steps. We use the AdamW optimizer (Loshchilov & Hutter, 2017) for pretraining, along with a cosine learning rate scheduler and a warm-up phase comprising 5% of the total training steps. To reduce GPU memory consumption, we adopt DeepSpeed optimization (Rasley et al., 2020), BF16 precision, and gradient checkpointing. All experiments are conducted on a cluster of H100-80G GPUs.

Table 8: Four navigation dataset statistics, including domain, the number of trajectories, and the average length of trajectory.

| Dataset | Domain | Trajs. | Len. |
|---------|--------|--------|------|
| AndroidControl | Mobile | 15,283 | 5.5 |
| GUI-Odyssey | Mobile | 7,735 | 15.4 |
| AITZ | Mobile & Web | 506 | 7.5 |
| Mind2Web | Web | 2,350 | 7.3 |

Instruction: Using Opera, find the date of the next Olympics opening ceremony and add a reminder for it in your Calendar.

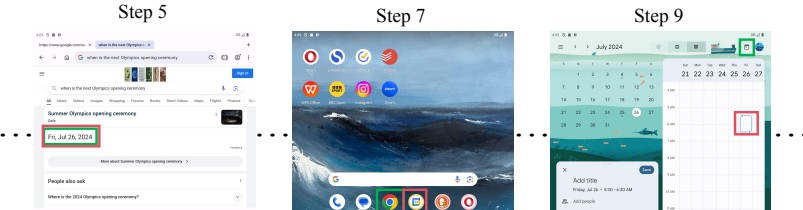

Figure 6: Qualitative results on the GUI-Odyssey dataset (Lu et al., 2024). High-level instructions are visualized at the top of each image. Predicted tap positions from our MTP (Qwen2.5-VL-3B) are shown in red, while those from the base model (Qwen2.5-VL-3B) are shown in green. Best viewed with zoom.

## D    QUALITATIVE ANALYSIS

We present a qualitative comparison on the GUI-Odyssey dataset (Lu et al., 2024) between the base model (Qwen2.5-VL-3B) and MTP (Qwen2.5-VL-3B) (Bai et al., 2025), focusing on their global planning capabilities. As shown in Figure 6, the task involves adding the date of the Olympics opening ceremony to the calendar. The base model exhibits two major planning failures, including opening Google Chrome instead of the calendar application in the step $7^{th}$ and selecting an incorrect date in the $9^{th}$. In contrast, MTP successfully executes the intended sequence of actions, demonstrating that masked trajectory pretraining significantly enhances global planning performance.

Moving from global to local planning, we present a qualitative comparison on the AndroidControl-Low dataset (Li et al., 2024) between the base model (Qwen2.5-VL-3B) and MTP (Qwen2.5-VL-3B), highlighting their differences in local planning capabilities through four representative examples. For instance, as shown in Figure 7 (a), MTP demonstrates stronger generalization during pretraining. While the base model tends to tap on "multiple options" when it fails to recognize the "send" icon, MTP correctly identifies and taps the intended target.

## E    ANALYSIS OF BENCHMARK STATISTICS

We evaluate the multi-step execution capabilities of MTP using four navigation datasets. Table 8 summarizes the statistics of AndroidControl (Li et al., 2024), GUI-Odyssey (Lu et al., 2024), AITZ (Zhang et al., 2024), and Mind2Web (Deng et al., 2023).

## F    THEORETICAL ANALYSIS

In this section, we analyze the consistency of training objectives in terms of optimization direction between direct unification mixture optimization and MTP from a gradient perspective. Specifically, we define the direct mixture optimization of the three pretraining paradigms, where the gradient for parameter updates is represented by $\nabla_\theta L_{\mathrm{mix}}$. Here, $\theta_{\mathrm{STP}}, \theta_{\mathrm{ACP}}, \theta_{\mathrm{MP}}$ denote the parameters for state-transition prediction, action-centric chain-of-thought (CoT) prediction, and multi-turn prediction,

Click on the send icon at the top    Click on the tools    Click on the idea pin option    Click on the Search Bar

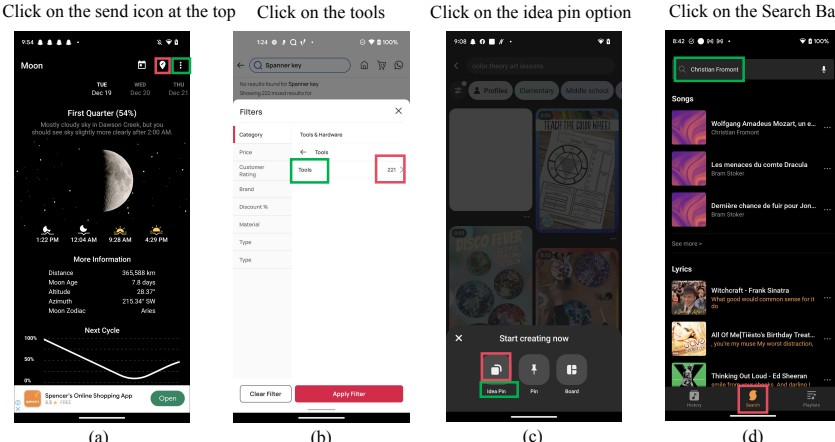

(a)       (b)       (c)       (d)

Figure 7: Qualitative results on the AndroidControl-Low dataset (Li et al., 2024). Low-level instructions are visualized at the top of each image. Predicted tap positions from our MTP (Qwen2.5-VL-3B) are shown in red, while those from the base model (Qwen2.5-VL-3B) are shown in green. Best viewed with zoom.

respectively. The total gradient is formulated as:

$$\nabla_\theta L_{\text{mix}} = \nabla_{\theta_{\text{STP}}} L_{\text{STP}} + \nabla_{\theta_{\text{ACP}}} L_{\text{ACP}} + \nabla_{\theta_{\text{MP}}} L_{\text{MP}}, \tag{6}$$

where $L_{\text{STP}}, L_{\text{ACP}}, L_{\text{MP}}$ are the loss functions of the three tasks. By summing the gradients from these tasks, the mixture optimization aims to balance the optimization directions of all paradigms.

The core idea of MTP is to treat any GUI trajectory $T$ as a whole and predict the masked components by randomly masking parts of the trajectory (such as $r_t$, $a_t$, or $m_t$). This approach unifies the dependency modeling of all pretraining tasks, as it allows each component of the trajectory to be predicted in the context of the entire sequence. From a gradient perspective, this means that the optimization process does not treat the tasks in isolation but instead optimizes the entire trajectory as a whole. By masking different components, MTP enforces a consistent training objective across tasks, ensuring that the gradients for different components of the trajectory align toward a common goal.

As mentioned in the main manuscript, we analyze the mean and variance of the gradient cosine similarity between MTP and the direct mixture of existing pretraining paradigms, with the detailed formulas provided in the appendix. The formula for the cosine similarity between two gradient vectors $\nabla A$ and $\nabla B$ is given by:

$$\cos(\nabla A, \nabla B) = \frac{\nabla A \cdot \nabla B}{\|\nabla A\| \, \|\nabla B\|} \tag{7}$$

where $\nabla A$ and $\nabla B$ can be any samples from the existing pretraining paradigms.

The mean of the gradient cosine similarity is defined as:

$$\mu_{\cos} = \mathbb{E}[\cos(\nabla A, \nabla B)] = \frac{1}{N} \sum_{i=1}^{N} \cos(\nabla A_i, \nabla B_i). \tag{8}$$

The variance of the gradient cosine similarity is given by:

$$\text{Varcos} = \mathbb{E}\left[ (\cos(\nabla A, \nabla B) - \mu\cos)^2 \right]. \tag{9}$$

## G    THE USE OF LARGE LANGUAGE MODELS (LLMs)

In this work, large language models (LLMs) were employed exclusively as general-purpose assistive tools to enhance the clarity, grammar, and readability of the manuscript. LLMs were not used

for research ideation, data analysis, model development, or any other aspect of scientific decision-making.

All scientific ideas, results, and conclusions presented in this paper were independently produced by the authors, who take full responsibility for the accuracy and integrity of the work, including any content refined with LLM assistance. No material generated by LLMs that could constitute plagiarism, fabrication, or scientific misconduct has been included.

