# OpenReview forum: "All in One: Unified Pretraining of GUI Agents via Masked Trajectory Prediction"
_ICLR.cc/2026/Conference — Submitted to ICLR 2026_

### Official Review · Reviewer_wjFB · 2025-10-17

**Soundness:** 3
**Presentation:** 2
**Contribution:** 3
**Rating:** 4
**Confidence:** 5

**Summary:**

This paper proposes MTP, a framework for the unified modeling of GUI data. By masking different components of GUI interaction trajectories and incorporating a role-aware adapter learning module, MTP unifies various training objectives and achieves state-of-the-art results on several benchmarks.

**Strengths:**

1. **Novel Unified Framework:** The paper introduces an innovative framework to uniformly model GUI Agent data. Through masked trajectory prediction and a role-aware adapter learning module, the model learns effectively from diverse types of training data.
2. **Clever Adapter Design:** The role-aware adapter learning module ingeniously trains multiple LoRA adapters to create a routing mechanism, which helps to further mitigate the negative impacts of data heterogeneity.

**Weaknesses:**

1. **Unified Modeling May Not Address Core Data Heterogeneity:** The idea of unified trajectory modeling is interesting but may not solve the fundamental problem of data heterogeneity. The authors classify GUI data into STP, ACP, and MP. While this classification is reasonable from a data format perspective, it doesn't delve into the essential capabilities of a GUI agent. A truly capable GUI agent requires domain knowledge, environmental awareness, and GUI reasoning. A training approach that starts from these core capabilities would be more meaningful than one that merely unifies data formats. A brute-force unification cannot resolve the skewed data distribution for each required capability, especially given that the distribution of open-source GUI agent data is severely imbalanced (e.g., abundant grounding data at 50-100M samples vs. scarce trajectory data around 100K). This introduces deeper issues related to data balancing and quality. For related perspectives, see the UI-Tars series[1,2] and OpenCUA[3].
2. **Misclassification of OS-Genesis Data:** Having thoroughly studied the OS-Genesis paper and its data, I am certain that its synthetic trajectories contain high-level instructions. According to this paper's own definitions, these should be classified as "semantically coherent trajectories." However, they are classified as "logically coherent trajectories," which is clearly incorrect.
3. **Lack of Online Benchmark Evaluation:** The experiments are missing an evaluation on online benchmarks, such as AndroidWorld. A GUI agent's true utility can only be proven by its performance in real-world, dynamic environments.
4. **Typo in Figure 2:** There is a drawing error in Figure 2. In the text portion, the components in $a_2m_3[Mask]$ are not separated by commas.

[1] Qin Y, Ye Y, Fang J, et al. Ui-tars: Pioneering automated gui interaction with native agents[J]. arXiv preprint arXiv:2501.12326, 2025.

[2] Wang H, Zou H, Song H, et al. Ui-tars-2 technical report: Advancing gui agent with multi-turn reinforcement learning[J]. arXiv preprint arXiv:2509.02544, 2025.

[3] Wang X, Wang B, Lu D, et al. Opencua: Open foundations for computer-use agents[J]. arXiv preprint arXiv:2508.09123, 2025.

**Questions:**

1. **Cross-Platform Action Space:** I am puzzled that the UniTraj dataset is said to span "five major operating systems," yet the action space is meticulously designed for Android. I understand some actions can be generalized, but how can the model cover the desktop action space without fundamental actions like `click`, `rightClick`, and `scroll`?
2. **Benchmark Coverage:** Following the previous question, if the data covers five operating systems, the evaluation should also include benchmarks from these different platforms. The current benchmarks are almost all for Android (AndroidControl, GUI-Odyssey, AITZ), with only one for the web (Mind2Web). There is a complete lack of desktop benchmarks. Perhaps the authors could evaluate on an offline desktop benchmark like AGENTNETBENCH[1]? This would be more convenient than online benchmarks like OSWorld or WAA, although an online desktop evaluation would be a more powerful response.
3. **Gains from Role-Aware Adapter Learning:** My understanding is that the role-aware adapter learning module is intended to route heterogeneous data on top of the unified model to further mitigate performance degradation from data diversity. The idea is good, but the experimental results do not show a significant gain. What could be the potential reasons for this?
4. **Suggestions for Figure 2:** I suggest improving Figure 2 by explicitly illustrating the different trajectory types. The formulas can be removed from the figure as they are already present in the text. Additionally, making the visual representations for inst, m, r, and a more distinct would improve clarity.
5. **Clarification on "m":** Perhaps I missed it, but I could not find a clear definition for the component labeled 'm'. Is it intended to represent the screenshot?
6. **Clarification on Lines 264-266:** I do not fully understand the statement in lines 264-266: "trajectories can contain up to six components but often include only a subset." What do these six components specifically represent? Neither the text nor Figure 3 provides a clear explanation.

[1] Wang X, Wang B, Lu D, et al. Opencua: Open foundations for computer-use agents[J]. arXiv preprint arXiv:2508.09123, 2025.

---

> ### Author Response · Authors · 2025-12-01
> **Official Comment by Authors (1/2)**
>
> Thank you for acknowledging our work! We are pleased that you highlighted our manuscript's focus on the critical issues in GUI mixture pretraining and the innovative framework introduced by MTP. We also appreciate your constructive suggestions for enhancing the effectiveness of our work, which have encouraged us to further explore the direction of GUI agents pretraining. We are happy to provide further clarification below.
>
> > Q1: Unified Modeling May Not Address Core Data Heterogeneity
>
> We fully agree with the reviewer’s viewpoint that a truly capable GUI agent requires domain knowledge, environmental awareness, and GUI reasoning, and that the community currently faces an imbalanced data scale between abundant grounding data and trajectory data. Our work focuses specifically on **how to fully leverage these trajectory data** to enhance model capability, which we view as an emerging direction as more diverse trajectory corpora become available in the open-source community.
>
> We refer to the UI-TARS series and other GUI pretraining work that utilizes trajectory data. We categorize these pretraining manners as three types, each corresponding to a **fundamental ability derived from trajectory supervision.**
>
> - State Transition Prediction (STP):  fine-grained state change understanding across consecutive GUI screenshots.
> - Action-centric CoT Prediction (ACP): fine-grained state change understanding across consecutive GUI screenshots.
> - Multi-turn Prediction (MP): long-horizon and goal-directed decision-making.
>
> While current work and open-source datasets cover aspects of all three abilities, directly mixing these three paradigms leads to inconsistent objectives and data heterogeneity. Prior work, such as UI-TARS, addresses this issue through extensive data cleaning and manual annotation. In contrast, masked trajectory prediction (MTP) consolidates diverse pretraining strategies into a consistent training objective via a masking-based manner without any additional data filtering.
>
> > Q2: Misclassification of OS-Genesis Data
>
> **A2:** Thank you for pointing out this issue. This was indeed a typographical error in the manuscript. We will correct it by categorizing OS-Genesis under the semantically coherent trajectories.
>
> > Q3: Lack of Online Benchmark Evaluation
>
> **A3:** We further include ablation studies on the AndroidWorld benchmark [1], where MTP surpasses direct mixture pretraining by 2.61, respectively. All experiments use the same base model, Qwen2.5-VL-3B.
>
> | Method                                     | **AndroidWorld SR** |
> | ------------------------------------------ | ---------------- |
> | Qwen2.5-VL-3B + Direct Mixture Pretraining | 10.43            |
> | Qwen2.5-VL-3B + MTP                        | **13.04**        |
>
> > Q4: Typo in Figure 2
>
> **A4:** Thank you for pointing out this issue. This was indeed a typographical error in the manuscript. We will add commas between the elements of *a2m3[Mask]* in Figure 2.
>
> > Q5: Cross-Platform Action Space
>
> **A5:** We acknowledge that the desktop action space was incompletely reported in the supplementary material. The full set of desktop actions includes additional actions, such as doubleClick, rightClick, middleClick, tripleClick, dragTo, moveTo, hotkey, and pess. We add an additional computer evaluation benchmark AGENTNETBENCH [2], as shown table below, MTP also achieves a substantial advantage over direct mixture pretraining.
>
> | **Method**                                 | Type Acc  | Ground Acc | Step Acc  |
> | ------------------------------------------ | --------- | ---------- | --------- |
> | Qwen2.5-VL-3B                              | 60.96     | 47.75      | 45.45     |
> | Qwen2.5-VL-3B + Direct Mixture Pretraining | 70.53     | 63.90      | 62.43     |
> | Qwen2.5-VL-3B + MTP                        | **73.43** | **67.31**  | **65.48** |
>
> > Q6: Benchmark Coverage
>
> **A6:** Thank you for the helpful suggestion, which further strengthens the evidence supporting MTP by evaluating on AGENTNETBENCH [2].  MTP achieves a substantial performance advantage over direct mixture pretraining on the AGENTNETBENCH [2].

---

> ### Author Response · Authors · 2025-12-01
> **Official Comment by Authors (2/2)**
>
> > Q7: Gains from Role-Aware Adapter Learning:
>
> **A7:** The core purpose of the role-aware adapter learning module is not to deliver large accuracy gains, but rather to address the severe gradient conflicts and training instability caused by heterogeneous GUI data. This module mitigates this issue by routing different token roles to specialized low-rank adapters. While this mechanism leads to only moderate accuracy gains, its primary contribution lies in stabilizing training and aligning gradients across heterogeneous data sources. The most direct evidence is presented in Figure 4 of the manuscript, where this module notably improves gradient cosine similarity (mean +0.17) and reduces gradient variance (–0.03).
>
> > Q8: Suggestions for Figure 2.
>
> **A8:** Thank you for the helpful suggestions regarding Figure 2. We will change in the modified version.
>
> > Q9: Clarification on 'm'.
>
> **A9:** 'm' indicates the screenshot. We will make a clear clarification for the modified version.
>
> > Q10: Clarification on Lines 264-266.
>
> **A10:** The six components refer to high-level instruction, screenshot, observation, thought,  low-level instruction, and action. Different datasets provide different subsets of these components. For example, Aguvis [3], OpenCUA [2], and a portion of ScaleCUA [4] contain all six components in each sample. In contrast, TongUI [5] and the navigation portion of ScaleCUA [4] include only four components, including high-level instruction, screenshot, low-level instruction, and action.
>
> Reference:
>
> [1]  Rawles, Christopher, et al. "Androidworld: A dynamic benchmarking environment for autonomous agents." *arXiv preprint arXiv:2405.14573* (2024).
>
> [2] Wang, Xinyuan, et al. "Opencua: Open foundations for computer-use agents." *arXiv preprint arXiv:2508.09123* (2025).
>
> [3] Xu, Yiheng, et al. "Aguvis: Unified pure vision agents for autonomous gui interaction." *arXiv preprint arXiv:2412.04454* (2024).
>
> [4] Liu, Zhaoyang, et al. "Scalecua: Scaling open-source computer use agents with cross-platform data." *arXiv preprint arXiv:2509.15221* (2025).
>
> [5] Zhang, Bofei, et al. "TongUI: Building Generalized GUI Agents by Learning from Multimodal Web Tutorials." *arXiv preprint arXiv:2504.12679* (2025).

---

### Official Review · Reviewer_tHb9 · 2025-10-27

**Soundness:** 2
**Presentation:** 3
**Contribution:** 2
**Rating:** 2
**Confidence:** 3

**Summary:**

The paper proposes a new training paradigm which uses Masked Trajectory prediction to unify all the open-sourced datasets. It also proposes role-aware adapters to address the challenge of data heterogeneity.

**Strengths:**

- proposes masked trajectory prediction for a unified training paradigm
- gathers open-source datasets and process to a UniTraj dataset for the MTP training
- uses role-aware adapters for data heterogeneity

**Weaknesses:**

- The experiment result does not show strong improvement of MTP training. In Table 1, compared with direct mixture training, the performance gain of MTP is only <1%. In table 2, the improvement from base model Qwen2.5-VL is very likely due to the substantial training data, while the improvement is barely around 1.4%.
- The experiment results do not show the strong contribution or application of the "pretraining" as defined in the paper. In table 2 and 3, even though with the substantial training data, Qwen2-VL+MTP performance still falls behind MiMo-VL, Qwen2.5-VL base models. Besides, from my knowledge, UI-Genie with only 90K open-source trajectories can achieve 74.2 on AndroidControl-High and 94.3 on AndroidControl-low which is even higher than Qwen2.5-VL+MTP.
- The study lacks the experiments on dynamic evaluation system such as AndroidWorld or AndroidLab or OSWorld. Currently the latest researches all evaluate their agents on such systems which represent a more realistic evaluation, since the static benchmarks such as AndroidControl or Mind2Web has significant drawbacks.
- More agents should be compared or listed, such as UI-TARS-1.5, GUI-OWL, UI-Venus, UI-Genie, etc.

**Questions:**

- In Table 1, why 2 "step acc" columns are listed with different numbers? Might be typo

---

> ### Author Response · Authors · 2025-12-01
> **Official Comment by Authors**
>
> Thank you for acknowledging our work! We are glad that you highlighted MTP as a simple and effective framework for consolidating diverse pretraining strategies into a consistent training objective for GUI agents. We also appreciate your constructive questions, and we are happy to provide further clarification below!
>
> >Q1: Limited performance among the base model, direct mixture pretraining, and MTP in the table1 and table2.
>
> **A1:** In the manuscript, we compare the performance difference between MTP and direct mixture pretraining, where MTP outperforms direct mixture pretraining by 1.4 Step Acc (from 71.5 to 72.9) in Figure 1a and by 0.43 Step Acc (from 66.12 to 66.55) in Table 1.
>
> During the rebuttal period, we have incorporated newly available open-source GUI trajectory data, such as OpenCUA [1] and ScaleCUA [2], into the pretraining stage to further improve performance over both the base model and the direct mixture pretraining. Specifically, we conduct ablation studies on the AndroidControl-High dataset [3] using Qwen2.5-VL-3B as the base model, evaluating both zero-shot and SFT performance against direct mixture pretraining and the base model. Under the zero-shot setting, MTP surpasses the base model and direct mixture pretraining by **3.43**% and **2.30**% Step Acc, respectively. Under the SFT setting, MTP further exceeds the base model and direct mixture pretraining by **2.60**% and **1.42**% Step Acc, respectively.
>
> | Method                                    | Zero Shot on AndroidControl-High |            |           | SFT on AndroidControl-High |            |           |
> | ----------------------------------------- | -------------------------------- | ---------- | --------- | -------------------------- | ---------- | --------- |
> |                                           | Type Acc                         | Ground Acc | Step Acc  | Type Acc                   | Ground Acc | Step Acc  |
> | Qwen2.5VL-3B                              | 78.15                            | 68.50      | 57.35     | 85.90                      | 73.60      | 68.20     |
> | Qwen2.5VL-3B + Direct Mixture Pretraining | 79.38                            | 68.89      | 58.48     | 85.95                      | 75.71      | 69.38     |
> | Qwen2.5VL-3B + MTP                        | **80.15**                        | **69.61**  | **60.78** | **86.06**                  | **76.65**  | **70.80** |
>
> > Q2: Comparison between UI-Genie.
>
> **A2:** To the best of our knowledge, the UI-Genie manuscript [1] reports that “**458k samples** derived from existing datasets and **59k process-reward samples** collected during self-improvement cycles”, **rather than the 90k samples** mentioned by the reviewer. Although UI-Genie leverages a larger dataset and higher-quality trajectories than MTP, it achieves only **74.2**% on AndroidControl-High, which is lower than MTP’s **74.4**%, while UI-Genie’s is only higher than MTP on AndroidControl-Low.
>
> > Q3: Online evaluation.
>
> **A3:** We further include ablation studies on the AndroidWorld benchmark [2], where MTP surpasses direct mixture pretraining by 2.61, respectively. All experiments use the same base model, Qwen2.5-VL-3B.
>
> | Method                     | **AndroidWorld SR** |
> | -------------------------- | ---------------- |
> | Direct Mixture Pretraining | 10.43            |
> | MTP                        | **13.04**        |
>
> > Q4: More agents compared such as UI-TARS-1.5, GUI-OWL, UI-Venus, UI-Genie, etc.
>
> **A4:** UI-TARS-1.5, GUI-OWL, and UI-Venus are technical reports that heavily rely on in-house datasets and extensive data cleaning. While our focus is on addressing inconsistent optimization direction and data heterogeneity issues introduced by direct mixture training. MTP is trained solely on open-source trajectory data, and therefore, it does not benefit from the same data quality and quantity as the datasets used in the aforementioned technical reports. As a result, we believe this comparison is not entirely fair, and therefore, we have not included these works in the manuscript.
>
> > Q5: Typo error in table 1.
>
> **A5:** Thank you for pointing out this issue. This was indeed a typographical error in the manuscript. The first column should be Type Acc, and the third column should be Step Acc.
>
> Reference:
>
> [1] Xiao, Han, et al. "UI-Genie: A Self-Improving Approach for Iteratively Boosting MLLM-based Mobile GUI Agents." *arXiv preprint arXiv:2505.21496* (2025).
>
> [2] Rawles, Christopher, et al. "Androidworld: A dynamic benchmarking environment for autonomous agents." arXiv preprint arXiv:2405.14573 (2024).

---

### Official Review · Reviewer_c37J · 2025-10-30

**Soundness:** 3
**Presentation:** 2
**Contribution:** 2
**Rating:** 4
**Confidence:** 4

**Summary:**

This paper proposed a unified framework named Masked Trajectory Prediction. It consolidates diverse pretraining strategies into a consistent training objective through a masking-based manner. A role-aware adapter learning module is designed to dynamically route each token to an appropriate optimization path. The experimental results on four benchmarks show the effectiveness of the proposed method.

**Strengths:**

1. The proposed MTP establishes a consistent training objective for GUI agents. The experimental results on standard benchmarks show the effectiveness and generalization of MTP.

2. The proposed MTP is simple and effective.

**Weaknesses:**

1. While the proposed MTP outperforms direct unified mixture training, the performance gains are marginal. As detailed in Table 1, the improvements range from 0.07% to 0.82%. These minimal gains raise questions about the practical significance of the method.

2. Given the ablation study presented in Table 1, the performance gains observed in Table 2 are likely attributable to the training data rather than the MTP itself.

**Questions:**

The major concern of this paper is the limited performance gains. Please see the Weaknesses.

---

> ### Author Response · Authors · 2025-12-01
> **Official Comment by Authors**
>
> Thank you for acknowledging our work! We are pleased that you highlighted the strengths of our proposed MTP framework, particularly its ability to establish a consistent training objective for GUI agents and its simplicity and effectiveness. Your constructive feedback on the presentation and contribution aspects of our work has encouraged us to further refine these areas. We are happy to provide further clarification below.
>
> > Q1 and Q2: Limited performance among the base model, direct mixture pretraining, and MTP in table1 and table2.
>
> During the rebuttal period, we have incorporated newly available open-source GUI trajectory data, such as OpenCUA [1] and ScaleCUA [2], into the pretraining stage to further improve performance over both the base model and the direct mixture pretraining. Specifically, we conduct ablation studies on the AndroidControl-High dataset [3] using Qwen2.5-VL-3B as the base model, evaluating both zero-shot and SFT performance against direct mixture pretraining and the base model. Under the zero-shot setting, MTP surpasses the base model and direct mixture pretraining by **3.43**% and **2.30**% Step Acc, respectively. Under the SFT setting, MTP further exceeds the base model and direct mixture pretraining by **2.60**% and **1.42**% Step Acc, respectively.
>
> | Method                                     | Zero Shot on AndroidControl-High |            |           | SFT on AndroidControl-High |            |           |
> | ------------------------------------------ | -------------------------------- | ---------- | --------- | -------------------------- | ---------- | --------- |
> |                                            | Type Acc                         | Ground Acc | Step Acc  | Type Acc                   | Ground Acc | Step Acc  |
> | Qwen2.5-VL-3B                              | 78.15                            | 68.50      | 57.35     | 85.90                      | 73.60      | 68.20     |
> | Qwen2.5-VL-3B + Direct Mixture Pretraining | 79.38                            | 68.89      | 58.48     | 85.95                      | 75.71      | 69.38     |
> | Qwen2.5-VL-3B + MTP                        | **80.15**                        | **69.61**  | **60.78** | **86.06**                  | **76.65**  | **70.80** |
>
> Reference:
>
> [1] Wang, Xinyuan, et al. "Opencua: Open foundations for computer-use agents." *arXiv preprint arXiv:2508.09123* (2025).
>
> [2] Liu, Zhaoyang, et al. "Scalecua: Scaling open-source computer use agents with cross-platform data." *arXiv preprint arXiv:2509.15221* (2025).
>
> [3] Li, Wei, et al. "On the effects of data scale on computer control agents." *arXiv e-prints* (2024): arXiv-2406.

---

### Official Review · Reviewer_WRhu · 2025-11-02

**Soundness:** 2
**Presentation:** 3
**Contribution:** 2
**Rating:** 4
**Confidence:** 4

**Summary:**

This paper proposes to address the problem of inconsistent training objectives and data heterogeneity in GUI pretraining. It proposes a unified training framework, named masked trajectory prediction (MTP). Extensive experiments on four benchmarks varify the effectiveness.

**Strengths:**

1. This paper tackles a novel problem in GUI pretraining. It is interesting to me.
2. The contribution of MTP is novel.

**Weaknesses:**

1. The performance gains over four benchmarks are relatively limited. Also, it is unclear how does the model perform when simply supervised finetuned on UniTraj dataset? Additionally, how does MTP compare with RL-based algorithm, or can MTP incorporate with RL, since the supervised finetuning-style methods have obvious performance ceilings.
2. The evaluated benchmarks are not that challenging. How does MTP perform in online settings (e.g., AndroidWorld, OS-World) ?
3. Is MTP sensitive to the quality of training data ? Could MTP benefit from further data scaling ?

**Questions:**

See above

---

> ### Author Response · Authors · 2025-12-01
> **Official Comment by Authors (1/2)**
>
> Thank you for recognizing our work! We are pleased that you highlighted the critical issues arising in GUI mixture pretraining and the innovative framework in MTP.  We also noticed you have some constructive questions about our work, and we're happy to elaborate further below!
>
> > Q1a: Limited performance among the base model, simple supervised finetuning on UniTraj, and MTP.
>
> **A1:** In the manuscript, the simple supervised finetuning on UniTraj corresponds to what we term direct mixture pretraining, and we will clarify this explicitly in the revised version. We compare the performance difference between MTP and direct mixture pretraining in the manuscript, where MTP outperforms direct mixture pretraining by 1.4% Step Acc (from 71.5 to 72.9) in Figure 1a and by 0.43% Step Acc (from 66.12 to 66.55) in Table 1.
>
> During the rebuttal period, we have incorporated newly available open-source GUI trajectory data, such as OpenCUA [1] and ScaleCUA [2], into the pretraining stage to further improve performance over both the base model and the direct mixture pretraining. Specifically, we conduct ablation studies on the AndroidControl-High dataset [3] using Qwen2.5-VL-3B as the base model, evaluating both zero-shot and SFT performance against direct mixture pretraining and the base model. Under the zero-shot setting, MTP surpasses the base model and direct mixture pretraining by **3.43**% and **2.30**% Step Acc, respectively. Under the SFT setting, MTP further exceeds the base model and direct mixture pretraining by **2.60**% and **1.42**% Step Acc, respectively.
>
> | Method                                     | Zero Shot on AndroidControl-High |            |           | SFT on AndroidControl-High |            |           |
> | ------------------------------------------ | -------------------------------- | ---------- | --------- | -------------------------- | ---------- | --------- |
> |                                            | Type Acc                         | Ground Acc | Step Acc  | Type Acc                   | Ground Acc | Step Acc  |
> | Qwen2.5-VL-3B                              | 78.15                            | 68.50      | 57.35     | 85.9                       | 73.6       | 68.2      |
> | Qwen2.5-VL-3B + Direct Mixture Pretraining | 79.38                            | 68.89      | 58.48     | 85.95                      | 75.71      | 69.38     |
> | Qwen2.5-VL-3B + MTP                        | **80.15**                        | **69.61**  | **60.78** | **86.06**                  | **76.65**  | **70.80** |
>
> >  Q1b: Incorpration RL with MTP.
>
> As the reviewer mentioned, SFT has a performance ceiling compared to RL-based methods. Nevertheless, our MTP-SFT pipeline surpasses several well-known RL approaches, as shown below, demonstrating that MTP effectively unlocks the model’s potential during the pretraining stage. Moreover, MTP can be seamlessly integrated with any RL algorithm, and due to the time constraints of the rebuttal period, we will incorporate RL with MTP in the final version.
>
> | Method           | Step Acc |
> | ---------------- | -------- |
> | Ui-tars [4]      | 72.5     |
> | GUI-R1[5]        | 51.7     |
> | AgentCPM-GUI [6] | 69.2     |
> | BTL-UI [7]       | 69.2     |
> | MTP              | **74.4** |
>
> > Q2: Online evaluation.
>
> We further include ablation studies on the AndroidWorld benchmark [2], where MTP surpasses direct mixture pretraining by 2.61, respectively. All experiments use the same base model, Qwen2.5-VL-3B.
>
> | Method                                     | **AndroidWorld** |
> | ------------------------------------------ | ---------------- |
> | Qwen2.5-VL-3B + Direct Mixture Pretraining | 10.43            |
> | Qwen2.5-VL-3B + MTP                        | **13.04**        |

---

> > ### Author Response · Authors · 2025-12-01
> > **Official Comment by Authors (2/2)**
> >
> > > Q3a: The sensitivity to data quality.
> >
> > MTP does not rely on high-quality training data. It can be effectively trained on logically trajectories even without high-level instructions or contain low-quality images. In particular, some trajectories extracted from web tutorials include instructional visual artifacts, such as red circles, arrows, highlights, and other overlays.
> >
> > > Q3b: The scaling capability of MTP.
> >
> > We further incorporate partial opencua and scalecua data during pretraining, which yields additional performance improvements and demonstrates the stronger scaling capability of MTP. Specifically, we evaluate on the AndroidControl-High dataset [3] using Qwen2.5-VL-3B as the base model after SFT. The first row shows the base model Qwen2.5-VL-3B, the second row shows the ICLR submission results, and the third row shows the improved rebuttal results after adding more data.
> >
> > | Method                                | Pretrain Data                | Type Acc  | Ground Acc | Step Acc  |
> > | ------------------------------------- | ---------------------------- | --------- | ---------- | --------- |
> > | Qwen2.5-VL-3B                         | --                           | 85.90     | 73.60      | 68.20     |
> > | Qwen2.5-VL-3B + MTP (ICLR submission) | UniTraj                      | 86.40   | 74.40    | 69.70   |
> > | Qwen2.5-VL-3B+ MTP (ICLR rebuttal)    | UniTraj + OpenCUA + ScaleCUA | **86.06** | **76.65**  | **70.80** |
> >
> > Reference:
> >
> > [1] Wang, Xinyuan, et al. "Opencua: Open foundations for computer-use agents." *arXiv preprint arXiv:2508.09123* (2025).
> >
> > [2] Liu, Zhaoyang, et al. "Scalecua: Scaling open-source computer use agents with cross-platform data." *arXiv preprint arXiv:2509.15221* (2025).
> >
> > [3] Li, Wei, et al. "On the effects of data scale on computer control agents." *arXiv e-prints* (2024): arXiv-2406.
> >
> > [4] Qin, Yujia, et al. "Ui-tars: Pioneering automated gui interaction with native agents." *arXiv preprint arXiv:2501.12326* (2025).
> >
> > [5] Luo, Run, et al. "Gui-r1: A generalist r1-style vision-language action model for gui agents." *arXiv preprint arXiv:2504.10458* (2025).
> >
> > [6] Zhang, Zhong, et al. "AgentCPM-GUI: Building Mobile-Use Agents with Reinforcement Fine-Tuning." *arXiv preprint arXiv:2506.01391* (2025).
> >
> > [7] Zhang, Shaojie, et al. "Btl-ui: Blink-think-link reasoning model for gui agent." *arXiv preprint arXiv:2509.15566* (2025).

---

### Author Response · Authors · 2025-12-01
**Response to All Reviewers**

We sincerely thank the reviewers for their thoughtful and constructive feedback, which has been invaluable in improving our work. We are particularly encouraged by the reviewers' recognition that our work focuses on inconsistent optimization objectives and data heterogeneity arising in direct mixture GUI pretraining, as well as introduces masked trajectory prediction (MTP) consolidates diverse pretraining strategies into a consistent training objective via a masking-based manner.

In particular, we appreciate the reviewers’ specific acknowledgments:

- **`WRhu`**, and **`wjFB`** highlight that MTP directly addresses the **critical issues** arising in mixture pretraining, including inconsistent optimization objectives and data heterogeneity.
- **`WRhu`**, **`c37J`**, and **`wjFB`** further emphasize that MTP introduces an **innovative framework** capable of uniformly modeling heterogeneous GUI-agent data.

Based on the valuable feedback, we have addressed all concerns and added comprehensive details.

- **More Performance gain：** We incorporate additional open-source GUI trajectory data and general-domain data into the MTP pretraining stage, which yields further performance improvements over both the base model and direct mixture training, and demonstrates the stronger scaling capability of MTP. Specifically, we conduct ablation studies on the AndroidControl-High dataset [1] using Qwen2.5-VL-3B as the base model, evaluating both zero-shot and SFT performance against direct mixture pretraining and the base model in the following table. Under the zero-shot setting, MTP surpasses the base model and direct mixture pretraining by **3.43%** and **2.30%** Step Acc, respectively. Under the SFT setting, MTP further exceeds the base model and direct mixture pretraining by **2.60%** and **1.42%** Step Acc, respectively.

| Method                                     | Zero Shot on AndroidControl-High |            |           | SFT on AndroidControl-High |            |           |
| ------------------------------------------ | -------------------------------- | ---------- | --------- | -------------------------- | ---------- | --------- |
|                                            | Type Acc                         | Ground Acc | Step Acc  | Type Acc                   | Ground Acc | Step Acc  |
| Qwen2.5-VL-3B                              | 78.15                            | 68.50      | 57.35     | 85.90                       | 73.60       | 68.20      |
| Qwen2.5-VL-3B + Direct Mixture Pretraining | 79.38                            | 68.89      | 58.48     | 85.95                      | 75.71      | 69.38     |
| Qwen2.5-VL-3B + MTP                        | **80.15**                        | **69.61**  | **60.78** | **86.06**                  | **76.65**  | **70.80** |

- **Online evaluation:** We further include ablation studies on the AndroidWorld benchmark [2] in the following table, where MTP surpasses direct mixture pretraining by **2.61%**, respectively. All experiments use the same base model, Qwen2.5-VL-3B.

| Method                                     | **AndroidWorld** SR |
| ------------------------------------------ | ------------------- |
| Qwen2.5-VL-3B + Direct Mixture Pretraining | 10.43               |
| Qwen2.5-VL-3B + MTP                        | **13.04**           |

- **Computer benchmark evaluation:**  We also include additional ablation studies on the AGENTNETBENCH [3] in following table, where MTP outperforms the base model and direct mixture pretraining by **20.03%** and **3.05%** Step Acc, respectively.

| **Method**                                 | Type Acc  | Ground Acc | Step Acc  |
| ------------------------------------------ | --------- | ---------- | --------- |
| Qwen2.5-VL-3B                              | 60.96     | 47.75      | 45.45     |
| Qwen2.5-VL-3B + Direct Mixture Pretraining | 70.53     | 63.90      | 62.43     |
| Qwen2.5-VL-3B + MTP                        | **73.43** | **67.31**  | **65.48** |

[1] Li, Wei, et al. "On the effects of data scale on computer control agents." *arXiv e-prints* (2024): arXiv-2406.

[2] Rawles, Christopher, et al. "Androidworld: A dynamic benchmarking environment for autonomous agents." *arXiv preprint arXiv:2405.14573* (2024).

[3] Wang, Xinyuan, et al. "Opencua: Open foundations for computer-use agents." *arXiv preprint arXiv:2508.09123* (2025).

---

### Meta-Review · Area_Chair_wn8e · 2026-01-07

**Summary:**

1. The performance gains over strong baselines are consistently marginal, a concern raised by nearly all reviewers, casting doubt on the practical impact of MTP.
2. The evaluation lacks sufficient online benchmarks. While AndroidWorld was added in rebuttal, key environments such as AndroidLab and OSWorld remain absent.
3. Several strong and relevant baselines (e.g., UI-TARS-1.5, GUI-OWL, UI-Venus, UI-Genie) are not included in the comparisons.
4. The scaling behavior of MTP appears limited: increasing data scale does not yield commensurate performance improvements.

**Reviewer Concerns:**

### Concerns Addressed in Rebuttal

* Minor typos and presentation issues
* Cross-platform action space clarification

### Concerns Not Adequately Addressed

* Absolute performance improvements remain marginal
* Limited evidence of effective scaling behavior
* Absence of online evaluation benchmarks, notably OSWorld, despite being explicitly requested by multiple reviewers

**Reviewer Scores:**

* **Reviewer WRhu (initial: 4)**
  Unlikely to increase, as absolute gains remain small and OSWorld experiments were not provided.

* **Reviewer c37J (initial: 4)**
  Unlikely to increase due to the limited magnitude of reported improvements.

* **Reviewer tHb9 (initial: 2)**
  May increase to 4, but unlikely beyond that: while Weaknesses 2/4 were reasonably addressed, Weaknesses 1/3 remain unresolved.

* **Reviewer wjFB (initial: 4)**
  Unlikely to increase, as Weaknesses 1/3 remain salient.

---

### Decision · Program_Chairs · 2026-01-26

Reject